# Shedding Light on the Effects of Calorie Restriction and Its Mimetics on Skin Biology

**DOI:** 10.3390/nu12051529

**Published:** 2020-05-24

**Authors:** Yeon Ja Choi

**Affiliations:** Department of Biopharmaceutical Engineering, Division of Chemistry and Biotechnology, Dongguk University, Gyeongju 38066, Korea; yjchoi@dongguk.ac.kr; Tel.: +82-54-770-2223

**Keywords:** skin aging, calorie restriction, intermittent fasting, CR mimetic, photoaging, skin appendages

## Abstract

During the aging process of an organism, the skin gradually loses its structural and functional characteristics. The skin becomes more fragile and vulnerable to damage, which may contribute to age-related diseases and even death. Skin aging is aggravated by the fact that the skin is in direct contact with extrinsic factors, such as ultraviolet irradiation. While calorie restriction (CR) is the most effective intervention to extend the lifespan of organisms and prevent age-related disorders, its effects on cutaneous aging and disorders are poorly understood. This review discusses the effects of CR and its alternative dietary intake on skin biology, with a focus on skin aging. CR structurally and functionally affects most of the skin and has been reported to rescue both age-related and photo-induced changes. The anti-inflammatory, anti-oxidative, stem cell maintenance, and metabolic activities of CR contribute to its beneficial effects on the skin. To the best of the author’s knowledge, the effects of fasting or a specific nutrient-restricted diet on skin aging have not been evaluated; these strategies offer benefits in wound healing and inflammatory skin diseases. In addition, well-known CR mimetics, including resveratrol, metformin, rapamycin, and peroxisome proliferator-activated receptor agonists, show CR-like prevention against skin aging. An overview of the role of CR in skin biology will provide valuable insights that would eventually lead to improvements in skin health.

## 1. Introduction

The skin is the largest organ of the body and provides important protection from life-threatening environmental factors. The skin undergoes physiological and functional deterioration as organisms age, which manifests as visible changes that are clearly apparent. The esthetic implication of skin aging has been a motivating factor behind numerous studies investigating this phenomenon. In addition, skin aging is largely influenced by extrinsic factors owing to its location; this process is called extrinsic skin aging, which is a separate process to intrinsic and chronological skin aging.

Calorie restriction (CR) is the most effective intervention to extend the lifespan of various organisms and has been used as a benchmark for longevity in anti-aging research [1,2]. There are numerous reports demonstrating the preventive effects of CR on aging and associated diseases, such as chronic nephropathies, cardiomyopathies, diabetes, autoimmune conditions, respiratory diseases, and neurological degeneration [3,4]. However, the effects of CR on the skin are poorly understood. The lifespan-prolonging changes induced by CR in the skin are believed to be less pronounced than in other major organs, including the liver, heart, and brain. This review serves to summarize the characteristics of skin aging and discuss current studies focusing on the effects of CR and alternative approaches to CR on cutaneous physiology and aging.

## 2. Skin Aging

### 2.1. Extrinsic Factors of Skin Aging

The aging process involves physiological and functional deterioration that progresses throughout the lifetime of an organism, induced by various genetic and non-genetic environmental factors. Eventually, homeostasis is disrupted, and susceptibility to disease or death is increased [5]. Skin aging is affected by both intrinsic and extrinsic factors, with extrinsic environmental factors contributing to the effects of chronological aging.

The concept of “exposome” was developed by American cancer epidemiologist Christopher Wild in 2005 [6] and refers to the totality of exposures to which an individual is subjected to from conception to death. The skin aging exposome, proposed by Krutmann et al., suitably describes the external and internal factors, and their interactions, that affect humans from conception to death, as well as the response of the human body to these factors that lead to skin aging [7]. The environmental factors of a skin aging exposome can be categorized into solar radiation (ultraviolet (UV) radiation, visible light, and infrared radiation), air pollution, tobacco smoke, nutrition, cosmetic products, and miscellaneous factors [7].

Among the extrinsic factors, UV radiation is an established accelerator of skin aging, through a process termed photoaging [8]. While UVC (200–280 nm) is filtered out by the ozone layer, UVB (280–320 nm) and UVA (320–400 nm) are the principal wavebands responsible for photoaging. An organism is exposed to sunlight UV comprising approximately 5% UVB and 95% UVA, but the degree of UVB exposure is expected to increase with ozone layer depletion [9]. UVB generally induces DNA damage, which results in skin tumorigenesis and causes excess melanin production and sunburn at high doses. UVA causes little DNA damage but generates significant oxidative stress, which mediates oxidative damage to DNA and non-DNA targets [10]. UVB and UVA both contribute to the characteristic features of photoaging [11].

Unlike experimental animals, humans who have lived strictly indoors for a lifetime are rare. In humans, photoaging is superimposed onto intrinsic aging. However, the difference between intrinsic aging and photoaging can be evaluated by comparing the features of UV-exposed skin sites, such as the face and dorsal side of the forearm, with non-exposed sites, e.g., the buttock skin. For example, while the loss of extracellular matrix (ECM) is a distinct feature of intrinsically aged skin, photoaged skin contains abundant elastin and collagen fibers, which are fragmented and disorganized [11,12]. Ultimately, the morphological changes and functional loss of skin during the aging process result from a combination of intrinsic and extrinsic aging.

### 2.2. Structural and Functional Alterations in Aged Skin

The skin is a complex organ with multiple cell types. It consists of two primary layers, the epidermis and dermis, as well as skin appendages, including hair follicles, sebaceous glands, sweat glands, and nails [13]. The physiological and functional alterations of each part of the skin during aging are reviewed in the following sections.

#### 2.2.1. Epidermal Changes in Aged Skin

The outermost layer of the skin, a highly specialized and multilayered epithelium, is called the epidermis. The mature epidermis is a stratified squamous epithelium that is composed of numerous keratinocyte layers, including the “stratum basale” (or the basal layer), “stratum spinosum” (or the spinous layer), “stratum granulosum” (or the granular layer), and the “stratum corneum” (or the corneal layer). Keratinocytes proliferate symmetrically and asymmetrically and differentiate, slowly moving towards the surface replacing old cells. During the terminal differentiation process, the cells become more flattened and water impermeable.

Changes in the thickness of the epidermis with age are varied, but mostly appear to be sustained or decreasing thickness. While the epidermis on the upper inner arm was reported to be thinner in the elderly [14], highlighting the alteration of the epidermis during intrinsic skin aging, other reports showed that epidermal thickness was not correlated with age [15,16,17]. In a murine study, older animals had a thinner epidermis at four different skin areas (dorsal, ventral, pinna, and footpad) of laboratory-raised CBA mice [18], but no differences were observed in the thickness of the dorsal and ventral epidermis of C57BL/6 mice [19]. In another study, photoaged skin appeared hypertrophic, atrophic, or unaltered, and histologically, the stratum corneum showed hyperkeratosis [8]. Cigarette smoking, another extrinsic factor of skin aging, was found to be negatively correlated with the thickness of the stratum corneum [16].

A vital function of the epidermis is to act as a protective interface between the body and the external environment, by preventing infection and the loss of body fluids, resisting mechanical stress, and participating in immune responses. The skin barrier function is also partially influenced by age. In menopausal women, measurements of transepidermal water loss showed a minor change in the hydration of the cornified layer, decreased sebum production, and a significantly higher skin surface pH [20]. Chronic itching is also common in aging skin. This may be due to the age-related decline in Merkel cell numbers, which act as mechanoreceptors [21], which causes the sense of touch to turn to an itching sensation [22].

A basement membrane lies beneath the epidermis at the dermo-epidermal junction, which adheres the epidermis to the dermis through a connection of the basal keratinocytes to the basement membrane by hemidesmosomes, and the fibroblasts in the dermis attach to the basement membrane by anchoring fibrils [23,24]. The junction is undulating. The downward folds of the epidermis are called epidermal ridges or rete ridges, and the upward projections of the dermis are called the dermal papillae. Significant flattening of the rete ridges was consistently observed in independent studies of different areas of skin [17,25]. This change is believed to reduce the interface between the epidermis and dermis, decreasing epidermal resistance to shearing stress, which makes the epidermis more fragile [26].

#### 2.2.2. Dermal and Hypodermal Changes in Aged Skin

The dermis is comprised of a connective tissue layer of mesenchymal origin and is subdivided into the following three layers, in order of proximity to the epidermis: papillary, reticular, and hypodermis. Fibroblasts, the most representative cells resident in the dermis, produce and secrete ECM proteins. Collagen is an essential ECM protein found in the dermis; type I and type III collagen are particularly abundant there. Other dermal ECM components include elastin fibers, proteoglycans, and hyaluronic acid, which provide strength, support, and flexibility.

Structurally, the dermis in humans and mice becomes thinner and loses elasticity with age [15,19,27]. Solar elastosis is a hallmark of human photoaging, characterized by an accumulation of partially degraded elastin fibers in the upper dermis. Although this alteration is not typically observed in intrinsic skin aging, an abnormal elastin network is also detected with age in sun-protected skin [8,26]. Massive deposition of other components of the ECM, such as glycosaminoglycans and interstitial collagen, has been observed in photoaged skin. The number of dermal fibroblasts and their capability to produce ECM are lower in intrinsically aged skin. Fibril collagen is degraded markedly with age, as well as by UV irradiation [28].

Matrix metalloproteinases (MMPs), a class of proteolytic enzymes, are considered the leading physiological cause of the breakdown of dermal ECM proteins. The expression and activities of MMPs are increased in aged skin and senescent fibroblasts, whereas the expression of tissue inhibitor of metalloproteinases (TIMPs) is decreased [29]. UVB irradiation also contributes to the activation MMPs [30]. Cathepsin K is a lysosomal protease that plays a vital role in clearing elastin that has been partially degraded by MMPs in the ECM [31]. Codriansky et al. reported that Cathepsin K was induced in young dermal fibroblasts as a response to UVA irradiation but not in fibroblasts from old donors [32].

Two interesting studies recently reported the cellular and molecular mechanisms underlying age-related dermal functions. Aged upper dermal fibroblasts gradually acquire the characteristics of the lower dermis, with reduced expression of ECM proteins and increased adipogenic traits [33]. Marsh et al. revealed that fibroblast positions are stably maintained over time, but clusters of fibroblasts are lost, and the membrane extends to fill the space of lost neighboring fibroblasts in aged skin. These findings provide a mechanism for a loss of cellularity in aged fibroblasts [34].

#### 2.2.3. Changes in Hair Follicles during Skin Aging

Hair follicles are comprised of an outer root sheath (ORS) and inner root sheath (IRS), which enclose the hair shaft. The hair bulb is at the base of the follicle, which contains proliferating matrix cells that grow to form the hair shaft and surround the dermal papilla at the bottom of the hair follicle. The dermal papilla consists of specialized mesenchymal cells [35]. The hair bulge, part of the ORS and located at the insertion site of the erector pili muscle of the hair follicle, is where epidermal stem cells reside. In adult mammals, hair grows in a regenerative cycle of phases, namely anagen (growth phase), catagen (regression phase), telogen (resting phase), and exogen (hair shaft shedding phase) [36]; this cycle is tightly regulated by the integrated action of multiple signaling pathways.

Elderly hairs become thinner, weaker, dry, dull, and sparse, due to hair follicle miniaturization and hair shaft weathering. Senescent alopecia is prevalent in the aged population, which is a diffuse and non-patterned type of hair loss that differs from androgenetic alopecia [37]. A lack of correlation between age and the total follicle number has been reported [38,39]. Old C57/Bl6 mice exhibited swelling hair follicles and a variable loss of normal hair follicle triplet patterning compared to young animals [40]. Furthermore, increased levels of inhibitors of Wnt, an activator of hair growth, such as dickkopf Wnt signaling pathway inhibitor 1 (DKK1) and secreted frizzled-related protein 4 (Sfrp4), were found in aged mice [41]. Aged hair follicle stem cells are believed to be a major contributor to a slow hair cycle and loss of hair during skin aging, which will be discussed in depth in Section 3.

Gray hair is one of the most noticeable signs of aging [37]. The number of melanocytes in the hair matrix decreases in aged hair follicles [42]. Melanocyte stem cells (MSCs) are maintained in the hair bulge area, and the mature melanocytes reside in the hair bulb. The frequency of melanocyte-inducing transcription factor (MITF)-positive melanocytes per basal keratinocytes in the hair bulge decreases significantly with age [43]. The abnormal maintenance of MSCs, together with a loss of differentiated progeny, contributes to physiological hair graying [43].

#### 2.2.4. Changes in Sweat Glands during Skin Aging

The sweat glands are small tubular structures in the skin, producing and excreting sweat. Functionally, sweat glands remove excess micronutrients, metabolic waste, and toxins from the body, and are involved in thermoregulation. Sweat glands can be divided into three types: eccrine, apocrine, and apoeccrine [44]. The eccrine sweat glands are the most numerous, distributed across almost the entire body surface area, and are smaller than the other two glands. Apocrine and apoeccrine are limited to specific regions of the body, such as axilla.

A decrease in the number of eccrine sweat glands and a shrunken morphology was detected in the scalp skin of old males (83.8 ± 2.8 years old) compared to younger males (33 ± 6.3 years old) [15]. The responsiveness of the eccrine sweat gland to pharmacological stimuli was estimated in different age groups of men. The results revealed a comparable density of activated glands but a lower sweat gland output per active gland in the old group (age >58 years old) [45], which implies a functional decline in the sweat glands during aging. Recent data suggested that epithelial autophagy contributes to the homeostasis of sweat glands, showing a significant decrease in the number of functional sweat glands in conditionally lacking Atg7 in K14-positive precursor cells [46]. In another study, the age-related reduction of sweat gland function was found to be regionally different [47]. In addition, sweating between old and young adults during exercise in the heat was comparable, indicating that the ability to regulate body core temperature during heat stress was retained in older adults [44].

#### 2.2.5. Changes in Sebaceous Glands during Skin Aging

Sebaceous glands are unique microscopic gland structures that accompany hair follicles. In humans, sebaceous glands are distributed throughout all skin sites but show a high abundance on the face and scalp. These glands secrete a complex oily and waxy mixture called sebum, which lubricates and waterproofs the skin and hair [48] and also participates in the immunity of mammals through the production of antimicrobial peptides, cytokines, and chemokines, such as interleukin (IL)-1β, IL-6, IL-8/CXCL-8, and tumor necrosis factors (TNFs) [49]. Sebum is comprised of triglycerides, wax esters, cholesterol esters, squalene, and free fatty acids [48]. Sebocytes are the major cells within the sebaceous glands. Fully mature sebocytes act in a holocrine manner; this is a unique secretion process that destroys the cell and results in the secretion of the product into the lumen [50].

During skin aging, the size and secretory activity of sebocytes decrease, which results in a decreased level of the surface lipid and dry skin [51]. Aged sebocytes were found to express more growth-regulated protein alpha (GRO-α), a CXC chemokine, which was attributed to the constitutive activation of NF-κB [52]. Cigarette smoke was found to decrease the level of scavenger receptor B1 (SRB1) [53], which is an oxidative stress-sensitive, transmembrane receptor that is well known for cholesterol uptake from high-density lipoprotein (HDL) [54]. Reduced SRB1 levels due to cigarette smoking compromised the cholesterol uptake of sebocytes, leading to an alteration of the sebocyte lipid content [53]. Moreover, age-related hormonal changes contribute to decreased lipid synthesis and changes in the gene expression profiles of sebocytes [55].

## 3. Effects of CR on Skin Aging

### 3.1. Effects of CR on Wound Healing

CR is universally believed to be a remarkable dietary manipulation of aging and age-related diseases, but its effects on the skin are poorly understood. Although CR has beneficial effects in other organs, it had an insignificant influence on age-related cutaneous phenotypes and an association with adverse outcomes in wound healing. Previous studies showed that CR retarded wound healing and collagen production [56,57,58,59]. During wound healing, activated fibroblasts transform to myofibroblasts and migrate to the area of the lesion, where they assist in closing the wound by promoting the synthesis and secretion of collagen. In different studies, the capacity of wound repair in animals with food ad libitum (AL), CR, and CR followed by refeeding for one month prior to wounding was compared [60,61]. Slower wound healing was observed in AL and CR aged animals compared to young subjects, which is consistent with previous data, but it was reported that CR animals with refeeding before the wound healed showed similar healing to that of the young animals, with enhanced synthesis of type I collagen. CR reduced collagen glycation, which are abnormal protein adducts detected in diabetic or aged skin, in older Rhesus monkeys by 30% [59]. These studies suggested that CR assisted in preserving the proliferative capacity required for wound repair.

### 3.2. Effects of CR on Morphological and Structural Changes in the Skin

A recent study on female 8-week-old Swiss mice fed a 60% reduced diet for six months revealed a thicker epidermis and reduced dermal white adipose tissue, demonstrating that tissue ultrastructure is modified by prolonged CR [62]. In addition, CR induced dermal vasculature development, accompanied by higher levels of vascular endothelial growth factor (VEGF), compared to AL mice [62,63]. Abdominal skin from 4-, 12-, and 24-month-old Fisher male rats that were fed a CR diet showed that age-related increase in the thickness of dermis and hypodermis was rescued. In contrast to previous data which showed thinner or age-independent changes in aged skin, the epidermis layer increased according to age and was comparable with AL and CR animals [63]. Interestingly, CR induced morphological changes to the fur coats of laboratory rats [62]. CR animals displayed significantly more and longer guard hairs in their skin fur coats than AL animals, whereas other types of hairs (Awl, Auchene, and Zigzag) remained unchanged. These changes provided a fur coat with better thermoregulatory properties. This same study also showed higher hair follicle growth in CR animals, which is associated with an increase in interfollicular and hair follicle stem cells.

A study that investigated the effect of CR on photoaging showed that CR reduced wrinkle formation when compared to AL animals, both with and without UVB irradiation [64]. Epidermal thickness increased after UVB radiation, which was accelerated by CR, as observed by corresponding epidermal proliferating cell nuclear antigen (PCNA) levels. CR also influenced the histological alteration upon UVB radiation, but further molecular and mechanistic evaluations are required to determine the precise effect of CR on photoaging.

### 3.3. Effects of CR on Skin Stem Cells

Changes in stem cells have been implicated primarily in aging, as well as skin aging, because adult stem cells in tissues are essential for organ homeostasis and repair. In the epidermis, a range of stem cell populations are located in different regions, and each stem cell compartment produces a subset of differentiated epidermal cells. Interfollicular epidermal (IFE) stem cells are localized in the basal layer of the epidermis, and hair follicle stem cells (HFSCs) and MSCs are in the bulge and the hair germ [65].

Age-related decline in the renewal capacity of the hair cycle fully involves HFSC aging. DNA damage accumulates in the HFSCs during repetitive hair cycling, which leads to proteolysis of COL17A1, an important component of the follicle stem cell niche [66]. Deficiency in COL17A1 results in HFSC loss of stemness and differentiation into an epidermal lineage. A comparison between young and aged murine epidermal stem cells (ESCs) showed that they have similar in vitro growth and differentiation potentials, but local environmental factors influence skin aging [67]. UV has been demonstrated to induce stem cell apoptosis in the basal layer and hair bulge, which contributed partially to epidermal atrophy, slow wound healing, and depigmentation. Intrinsically aged murine skin had a comparable abundance of CD34^+^ epidermal stem cells [40].

The anti-aging capability of CR is related to its ability to reprogram stemness and boost the regenerative capacity of stem cells. Previously, CR improved the functioning of various stem cell populations, including hematopoietic and intestinal stem cells in mice and germline stem cells in flies [68]. However, there is limited research investigating the effect of CR on skin-residing stem cells. One study showed that CR expanded pools of IFE stem cells and HFSCs in CR animals, which promoted the growth and maintenance of their fur coats [62]. Furthermore, stem cells are under the control of a rhythmic circadian machinery; CR reversed the reprogrammed daily rhythms to adapt to tissue-specific stress in aged epidermal stem cells [69].

### 3.4. Effects of CR on Carcinogenesis

CR, by the restriction of fats or carbohydrates, delayed the rate and reduced the incidence of papilloma development [70]. Additionally, CR prevented UV-mediated skin tumor formation [71]. CR decreased the expression of oncogenic H-Ras and significantly activated Ras-GTP in skin stimulated with 12-0-tetradecanoylphorbol-13-acetate (TPA) [72]. In addition, the TPA-induced activation of PI3K/Akt and p42/p44-MAPK signaling was reduced in CR skin. Chemically induced ulcerative skin was observed to be more infrequent in CR than AL skin. Furthermore, the decrease in p53 gene expression in p53^+/–^ mice may have reduced the beneficial effects of CR in these circumstances [73]. Together, these results suggest that CR prevents skin carcinogenesis.

### 3.5. Metabolic Effect of CR on Skin Aging

Molecular alterations to metabolically adapt to limited calorie intake mediates the beneficial effects of CR. CR stimulates respiratory rates by enhancing mitochondrial biogenesis and stimulating uncoupling between oxygen consumption and oxidative phosphorylation. A metabolic shift to a more oxidative phenotype has been reported in the dermal compartment [62]. Metabolomic analysis revealed that UV exposure induced catabolism of biomolecules and increased oxidative stress [74]. The metabolome data showed altered activity in upper glycolysis and glycerolipid biosynthesis and decreased protein and polyamine biosynthesis in aged skin [75]. CR-mediated metabolic alterations might be employed through changes in cellular signaling, epidermal barrier function, and skin structure during skin aging; further investigation is necessary to better determine the underlying molecular activities.

## 4. Effects of Alternative Ways of Dietary Restriction on Skin Aging

The food intake of experimental animals under CR in aging research is severely restricted; overall, calorie intake or food intake in CR models is reduced by approximately 10%–50%, without malnutrition, compared to AL controls [76], which could be challenging for humans to practice and sustain. Therefore, several practical approaches, such as intermittent (e.g., alternate day fasting) and periodic (fasting that lasts three days or longer, every two or more weeks) fasting, or alternative methods of dietary restriction, have been suggested for humans [77]. The restriction of specific nutrients rather than the decrease in total food intake has shown beneficial effects on lifespan extension and prevention of age-related diseases. In addition, various pharmacological interventions, from natural products to synthetic compounds, have been developed and studied to mimic the benefits of CR as an anti-aging strategy. However, the effects of these alternative ways of CR on skin biology, including the skin aging process and skin disorders, have been paid less attention. The following sections of this review describe current research into the effects of alternative dietary restriction approaches and CR mimetics on skin biology and aging.

### 4.1. Effects of Fasting on Skin Biology

A recent review summarized current literature on the impact of fasting on skin biology [78]. Most of the study focused on the efficacy of fasting on wound healing. While fasting for three days delayed wound healing [78], short-term, repeated fasting (four consecutive days, every two weeks) for two months before the wound, improved wound healing with increases in epithelialization, contraction, healing, collagen levels, and hydroxyproline [58]. This is consistent with the increased capacity of wound repair in the animals of the caloric restricted-refed group. In another study, four days of a diet that mimics fasting (FMD) reduced severe ulcerating dermatitis in C57BL/6 mice, which indicates that FMD protects against inflammation and inflammation-associated skin lesions [79]. Bragazzi et al., the author of the recent review [78], emphasized the need for evidence-based and standardized protocols of fasting and qualitative improvement in research on fasting and skin.

### 4.2. Effects of Specific Macronutrient Restriction on Skin Biology

Previous studies have shown that a decrease in either dietary protein or sugar can reduce mortality and extend the life span of *Drosophila* [80] and mice [81], independently of the calorie intake. Furthermore, the reduced intake of specific essential amino acids, such as methionine, tryptophan, or branched-chain amino acids, had beneficial effects on delaying aging or improving health [82,83]. There has been little research on the impact of macronutrient restriction on skin biology and aging.

Protein restriction (PR, 0% kcal protein of total calorie) or methionine restriction (MR, 14% kcal protein containing 0.05% methionine) regimens were tested in the context of wound healing in normal and diabetic animals [84]. The mice preconditioned with PR for one week or MR for two weeks before surgery showed comparable wound healing to mice fed a complete diet. Under diabetic conditions, PR or MR improved perioperative glucose tolerance and perioperative hyperglycemia, without any impairment in wound healing. These results lessen the concerns of poor wound healing or susceptibility to infection associated with the typical CR method during surgery, suggesting the potential clinical application of these regimens.

On the other hand, a carbohydrate-restricted diet promotes skin senescence in senescence-accelerated prone mice. Histologically, the epidermis and dermis were thinner in the carbohydrate-restricted group, and cutaneous expression of the senescence markers p16 and p21 and lipid peroxidation was increased by long-term carbohydrate restriction [85,86]. Considering that this group also showed a significant progression of visible aging and decreased survival rate, the duration of nutrient restriction should be carefully considered.

## 5. Protective Effect of CR Mimetics on Skin Aging and Skin Disorders

CR mimetics have attracted considerable attention for many years because of their health-promoting effects [87]. Notwithstanding the limitations of some mimetics partially mimicking the effect of CR and the unclear mechanisms of action of CR mimetics, CR mimetics still have numerous advantages including convenience of application such as its potential use as a health food supplement [88,89,90]. In skin aging research, many pharmacological compounds have been studied as potential skin aging interventions. In this section, the current knowledge about several well-known CR mimicking compounds in terms of skin aging and disorders will be discussed (Figure 1).

### 5.1. Sirtuin and Resveratrol

The activation of sirtuins, which are nicotinamide dinucleotide (NAD^+^)-dependent deacetylases, has been reported to extend the lifespan of various organisms, including yeast, worms, fruit flies, and mice [91,92,93] and has been identified as a mediator of the beneficial effects of CR. More than 14,000 compounds that activate sirtuin have been identified [94]. Resveratrol (3,5,4′-trihydroxystilbene) was identified as the first potent activator of Sirtuin and has been studied extensively as a CR mimetic [95,96,97,98,99,100].

SIRT1, the mammalian homolog of yeast SIR2, is expressed ubiquitously throughout the skin. Immunohistochemical staining of elderly skin showed that the level of SIRT1 decreased and there was a steady reduction in the proliferation of dermal fibroblasts [101]. UVA and UVB irradiation also induced a decrease in gene expression or activity of SIRT1 in dermal fibroblasts [102,103], keratinocytes [104], and melanocytes. The overexpression of SIRT1 prevented human skin fibroblast senescence through deacetylation of forkhead box O3α (FOXO3α) and p53 [105]. Epidermis-specific SIRT1 deletion inhibited the regeneration of both the epidermis and dermal stroma, which shows that epidermal SIRT1 is essential for wound repair [106]. The strong correlation between skin aging and sirtuin expression and extensive mechanistic studies have supported SIRT1 as a pharmacological target and resveratrol as a powerful prevention therapy of skin aging.

The topical application of 2% resveratrol increased the repair of tissue wounds more so than for vehicle-treated rats and was associated with the induction of angiogenesis, fibroplasia, and collagen organization [107]. The administration of resveratrol to wounds improved epithelization, hair follicle regeneration, and collagen deposition in both young and old rodents [108]. Resveratrol stimulates the production of collagen types I and II, reduces the expression of AP-1 and NF-kB factors, and slows down the process of skin photoaging in human keratinocytes and mouse skin [109]. Oxidative stress-induced senescence was ameliorated by resveratrol in primary human keratinocytes [110]. Resveratrol was initially characterized as a SIRT activator, but other types of signaling, such as AMP-activated protein kinase (AMPK) and FOXO3, also contribute to its actions.

Resveratrol has been used increasingly in cosmetology and dermatology because of its antioxidant, anti-inflammatory, anti-proliferative, and anti-pigmentation properties [111]. Epidermal permeation of resveratrol has been assessed in vitro and in vivo. Most of the resveratrol was detected in the stratum corneum, and resveratrol penetrated the porcine skin at 20–49 μm, corresponding to the viable epidermis, at a constant concentration [112]. Dietary supplements and various types of cosmetics, such as sunscreen and ampules containing resveratrol for skin rejuvenation, are currently available, studies are underway to improve the delivery of the topical application of resveratrol to enhance its efficacy and stability [113,114,115,116].

### 5.2. AMPK and Metformin

Metformin (*N’,N’*-dimethylbiguanide) is one of the first-line drugs for treating type 2 diabetes [117]; its anti-aging property was illustrated by lifespan extension in *C. elegans* [118,119,120], *Drosophila* [121], and mice [122,123]. Although the precise molecular mechanisms of the effects of metformin remain unclear, it is known that metformin activates AMPK, which serves as an energy sensor and regulator of glucose homeostasis [124]. AMPK signaling intersects with the mammalian target of rapamycin (mTOR) [125], extracellular signal-regulated kinase (ERK) [126], and SIRT3 [127] and is also involved in mitochondrial biogenesis and activating autophagy [128].

AMPK activation in human skin reportedly decreases during the aging process [110]. Cutaneous AMPK activity is also downregulated by UVB irradiation in humans and mice [129]. AMPKα deletion in keratin 14-expressing ESCs resulted in hyperactive mTOR signaling leading to extensive hyperproliferation after acute wounding, UVB exposure, and phorbol ester application. These findings suggest that the essential role of ESC-specific AMPK is in the control of ESC proliferation and physiological skin repair [130]. Additionally, the activation of AMPK has a beneficial effect on oxidative stress-mediated UV-induced cellular senescence.

Both systemic and topical application of metformin successfully attenuated UVB-induced epidermal hyperplasia and skin tumorigenesis [129]. Metformin reversed the diminished collagen I production induced by UVA and suppressed MMP-1 expression, which substantiates the potential use of metformin in prevention against dermal aging. Metformin was reported to have a prominent effect on wound healing. Metformin-regulated AMPK/mTOR signaling resulted in M2 macrophage polarization by inhibiting NLRP3 inflammasome activation [131].

Zhao et al. estimated the efficacies of popular anti-aging agents, including resveratrol, metformin, and rapamycin; topical application of resveratrol and metformin, but not rapamycin, improved wound healing in young mice, and metformin exerted strong regenerative efficacy in aged skin [108]. Some of the beneficial effects of metformin have been achieved through cutaneous application. For example, a transdermal formulation for metformin, such as a cream and transdermal patch, was developed for patients who could not tolerate the oral dose or could not swallow large tablets. A recent paper reported a significant improvement in skin delivery by incorporating metformin into solid lipid nanoparticles and subsequently formulating an effective topical gel [132]. The future development of more advanced metformin delivery systems will allow for expanded application.

### 5.3. mTOR and Rapamycin

Rapamycin was first introduced as an inhibitor of mTOR, which is a serine-threonine kinase that regulates cell survival, growth, proliferation, motility, protein synthesis, transcription [133], and autophagy [134,135,136]. Rapamycin has also significantly increased the life span of various experimental models [137,138,139,140] and has potential as a CR mimetic. However, there has been limited research on the role of mTOR in skin aging. An age-related and UVB-induced increase in the activity of mTOR and RICTOR protein, which is a major component of mTOR complex 2 (mTORC2) [141], has been reported. In turn, the activation of mTORC2 signaling was found to mediate NF-kB activation during skin aging. The epidermal deficiency of mTORC2 signaling caused moderate tissue hypoplasia, reduced keratinocyte proliferation, and attenuated the hyperplastic response to TPA [142]. In an in vitro study, mTORC2 activity-deficient keratinocytes displayed a longer lifespan, less senescence, and an enhanced tolerance to cellular stressors. While this study was not performed in aged animals, it highlights the potential implications of mTOR signaling in skin aging and the therapeutic resistance of epithelial tumors. Moreover, mTOR has been implicated in the pathogenesis of various skin disorders, such as psoriasis [143,144], which strongly suggests the dermatological application of rapamycin.

Rapamycin effectively suppressed UVB-induced oxidative stress and collagen degradation in skin fibroblasts [145]. A recent clinical trial (ClinicalTrials.gov Identifier: NCT03103893) showed that topical rapamycin reduced the senescence and age-related features in human skin [146]. This study showed that the p16^INK4A^ level, a marker of cellular senescence, and solar elastosis was decreased in rapamycin-treated skin. Moreover, collagen VII, a critical component of the basement membrane, was increased and disorganized collagen was restored. These histological and molecular observations in rapamycin-treated skin highlight rapamycin as a potential anti-aging therapy with efficacy in humans.

### 5.4. PPAR Agonists

Peroxisome proliferator-activated receptors (PPARs) are nuclear receptors with diverse biological effects in the promotion of cellular proliferation and differentiation, lipid and carbohydrate metabolism, inflammatory responses, and tissue remodeling [147,148]. There are three PPAR isoforms, PPARα, PPARβ/δ, and PPARγ, which are distributed in different tissues and have selectivity and responsiveness to specific ligands [149]. Among them, PPARα and γ have been well investigated in aging research. The expression of PPARα and PPARγ genes was found to be decreased during aging, which was rescued by CR [147,150]. UV irradiation reduced PPARγ levels, which resulted in dysregulation in epidermal lipids in human skin, contributing to the development of skin photoaging [151].

PPARγ is also expressed throughout the skin and in most types of skin cells [149]. In the epidermis, PPARγ has an essential role in skin barrier regulation [152] and negatively regulates the gene expression of proinflammatory genes through the antagonization of inflammatory transcription factors NF-κB and AP-1. Therefore, synthetic ligand “glitazones,” which are a class of oral antidiabetic drugs, were applied; they had protective effects on inflammatory skin disorders, such as atopic dermatitis and psoriasis [153,154].

In addition to well-known agonists of PPARs, the beneficial effects of potential novel ligands in skin aging have been reported. Abietic acid, as a PPARα/γ dual ligand, decreased UVB-induced MMP-1 expression significantly by downregulating UVB-induced MAPK and NF-κB signaling in dermal fibroblasts [155]. Magnesium lithospermate B activated PPARβ/δ, leading to the upregulation of collagen expression in aged murine skin [156]. Treatment with the synthetic compound MHY966 (2-bromo-4-(5-chloro-benzo[d]thiazol-2-yl) phenol), a novel PPARα/γ dual agonist, protected UVB-exposed hairless mice from lipid peroxidation and elevated cutaneous proinflammatory mediators, including NF-κB, iNOS, and COX-2 [157].

## 6. Conclusions

Dietary restrictions affect the structure and function of skin. CR has beneficial effects on skin aging in terms of wound repair, stem maintenance, and carcinogenesis. While the physiological and pathological features of aged skin are relatively well characterized, the full effects of CR on skin physiology remain to be elucidated. Investigations pertaining to the effects of CR and CR alternatives on skin aging are limited, and extensive research is necessary to resolve these gaps in knowledge. Skin disease is increasing in prevalence among the elderly, and while many age-related skin disorders are not lethal, they are integral to general health status and overall quality of life. Currently, hormone therapy, antioxidant intervention, and the therapeutic application of stem cells are used to treat skin aging. The use of CR and CR mimetics has great potential to rejuvenate and maintain healthy skin, as well as improve age-related skin disorders. This review substantiates the need for further investigation into CR and related mimetics as potential therapeutic agents for skin aging and age-related disorders.

## Figures and Tables

**Figure 1 nutrients-12-01529-f001:**
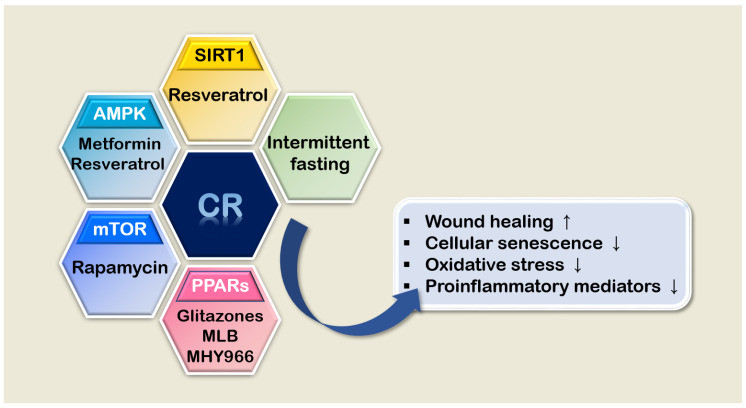
Summary of CR mimetics and their effects on skin aging. AMPK, AMP-activated protein kinase; CR, calorie restriction; MLB, magnesium lithospermate B; mTOR, the mammalian target of rapamycin; SIRT1, the mammalian homolog of SIR2; PPAR, peroxisome proliferator-activated receptors.

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
