# Peer review of "Shedding Light on the Effects of Calorie Restriction and Its Mimetics on Skin Biology"

_nutrients, 2020, doi:10.3390/nu12051529_

Round 1

Author Response

Thank you for the comments. All your comments were very helpful to improve the manuscript. Please see the attachment. 

Reviewer 2 Report

This review has been drastically improved.  The subject manner is more organized and flows better than the previous version.  Use of English is also good which was an issue with the previous version.  Much of the suggestions made by the reviewers were applied to the revision.

Author Response

Thank you for the comments. All your comments were very helpful to improve the manuscript.

This manuscript is a resubmission of an earlier submission. The following is a list of the peer review reports and author responses from that submission.

Round 1

Reviewer 1 Report

The review provided a general background of the structure of the skin and the consequences of aging and photoaging.  The next section tackles the influence of caloric restriction as a whole on skin and then goes more into studies on restricting different parts of the diet.  Finally the last section focused on different mimetics of caloric restriction and how they benefited the skin. 

The paper was structured well and followed a logical order.  It did well in making connections between different studies and being succinct.  The introduction needs work on the language as it does not quite flow, was choppy and has some grammatical errors.  The latter part of the paper seems to flow better where most of the content was contained.

The paper seems to lean in the direction of the metabolomic field.  Perhaps a reference on that may be added to strengthen the paper. 

Reviewer 2 Report

The structure of the article is good itself but there is a lack of balance in terms of content. Skin ageing should be reduced in order to have an article more focused on the CR benefits.

The conclusion should not be used to discussed additional articles.

English needs to be improved - a lot. I would like to suggest peers/colleague to further review the article prior submission.

Reviewer 3 Report

The review article “Shedding light on the effect of calorie restriction and its mimetics on skin biology” wants to give an overview about the past and current literature on the effects of calorie restriction on skin biology. The topic is appealing and the idea underneath can be of potential interest. Although the whole manuscript presents a general problem with the style and the structure, which does not allow the reader to follow the concepts and understand the messages clearly. I strongly suggest to carefully review the whole manuscript and think about a way to restructure it.

The whole part 2 (skin structure and functions) is way too long: it consists in a general description of the skin structure which seems not fitting well with the goal of the review. The verb tenses always switch from past to present. The conclusion is too concise and misses of future remarks about the topic. Many grammar mistakes. The English style needs to be reviewed and many sentences have to be rephrased. Here only some examples:

Page1, line 34: ……photoaging which, is ultraviolet (UV) irradiation-induced aging.

Page 1, 2nd paragraph: a reference is missing

Page 1, line 35

Page 1, line 43.

Page 2, line 44

Page 2, line 47: ….on the effect of CR on the cutaneous…

Page 2, line 57

Page 2, line 62

Page 2, 1st paragraph: there is no verb in the second sentence.

Page 2, title 2.2 should be dermis and hypodermis.

Page 2: example, correct with a phrase like:  The fibroblasts, the most representative cells resident in the dermis,….

Page 2, line 87:  These structures make the dermis provides functions.. 

Page 3, line 107: Sebaceous glands are a unique microscopic gland structure…

Page 3, improve form of 1st and 2nd paragraphs.

Part 3.1:  intrinsic ageing factors are not discussed.

Page 4, line 157

Page 4, par. 3.3

Page 5, line 222

Line 374: Metformin reported to has a prominent effect on…